# OpenReview forum: "Personality Alignment of Large Language Models"
_ICLR.cc/2025/Conference — ICLR 2025 Poster_

### Official Review · Reviewer_nZZi · 2024-10-27

**Soundness:** 3
**Presentation:** 3
**Contribution:** 2
**Rating:** 5
**Confidence:** 4

**Summary:**

This paper explores the personality alignment of large language models (LLMs). Specifically, it introduces a new dataset and proposes the PAS method for personality alignment based on this dataset.

**Strengths:**

The strengths of this paper can be summarized in three key points:

- It contributes a new dataset (or more accurately, a dataset generation pipeline).
- The proposed method is simple yet highly effective.
- The experimental analysis is comprehensive, with particularly detailed descriptions of the experimental setup.

**Weaknesses:**

However, I believe the paper has the following weaknesses:

- The scope of the contribution is somewhat limited. I would have liked to see this method applied to a broader range of personalities, rather than being restricted to just the five personalities of the Big Five model. The authors could consider additional datasets, such as the Dark Triad or even the MBTI test (though MBTI remains controversial in psychology). Expanding in this way would enhance the paper’s overall contribution.

- While the proposed method performs well on the dataset, it is actually quite similar to approaches used in many previous studies [1][2]. As a result, the novelty of the method is somewhat lacking, and the authors should be careful not to overstate their contribution.

- Essentially, the proposed method is a form of personalized alignment, so the authors should compare it against more baselines [3].

- Figures 6 and 7 show a significant disparity between LLM-as-a-judge evaluations and human evaluations, which makes me question the consistency and reliability of the judgment process.

- Other details: In line 1304, the authors mention that human annotators come from machine learning and computer science, but ML is a part of CS. Additionally, could the authors disclose the educational background of the annotators (undergraduate or graduate)?

[1] Zheng, Chujie, et al. "On prompt-driven safeguarding for large language models." *Forty-first International Conference on Machine Learning*. 2024.

[2] Wang, Haoran, and Kai Shu. "Backdoor activation attack: Attack large language models using activation steering for safety-alignment." *arXiv preprint arXiv:2311.09433* (2023).

[3] https://github.com/liyongqi2002/Awesome-Personalized-Alignment

**Questions:**

Could the authors explain why the PAS method is so effective by looking at the internal hidden layer representations of the model? Intuitively, direct training approaches like DPO/PPO might yield more noticeable improvements, but the results here contradict my expectations.

In line 429, the authors mention "Why Did Scaling Laws Fail?" but don't seem to fully answer this question. I would like the authors to explain this from the perspective of the domain's specificity. Is it because larger models learn more general alignment, which makes it harder for them to excel at aligning with a specific personality?

---

> ### Author Response · Authors · 2024-11-20
> **Response 1**
>
> >  The scope of the contribution is somewhat limited. I would have liked to see this method applied to a broader range of personalities, rather than being restricted to just the five personalities of the Big Five model. The authors could consider additional datasets, such as the Dark Triad or even the MBTI test (though MBTI remains controversial in psychology). Expanding in this way would enhance the paper’s overall contribution.
>
> Thank you for this insightful suggestion about expanding the scope of personality assessment. We have follow your advise and taken steps in this direction, **expanding our evaluation framework by incorporating the Dark Triad inventory alongside the Big Five traits.** Specifically, we collected and analyzed 18,192 independent samples measuring Machiavellianism, Narcissism, and Psychopathy through 27 targeted questions. Our results demonstrate PAS's effectiveness across these additional personality dimensions - achieving strong alignment scores for Machiavellianism (0.96), Narcissism (1.85), and Psychopathy (1.67) when using the Llama-3-8B-Instruct model. These scores are particularly noteworthy given the complexity of aligning models with these challenging personality traits. The Dark Triad provides an important complementary perspective to the Big Five, allowing us to evaluate alignment across both socially desirable and potentially problematic personality dimensions.
>
>
> **Table 1: Dark Triad**
>
> ###### GPT-4o Results
>
> | Method   | Machiavellianism | Narcissism | Psychopathy |
> | -------- | ---------------- | ---------- | ----------- |
> | Few-Shot | **0.80**         | **0.76**   | **0.83**    |
> | P^2      | 1.17             | 2.04       | 2.00        |
>
> ###### Llama-3-8B-Instruct Results
>
> | Method             | Machiavellianism | Narcissism | Psychopathy |
> | ------------------ | ---------------- | ---------- | ----------- |
> | PPO                | 1.48             | 1.98       | 2.19        |
> | DPO                | 1.41             | 1.99       | 2.12        |
> | Prompt-MORL        | 1.42             | 2.14       | 1.78        |
> | Personalized-Soups | 1.08             | **1.76**   | 1.84        |
> | Few-Shot           | 1.16             | 2.03       | 2.00        |
> | P^2                | 1.17             | 2.04       | 2.01        |
> | PAS (Ours)         | **0.96**         | 1.85       | **1.67**    |
>
> ###### Llama-3-70B-Instruct Results
>
> | Method             | Machiavellianism | Narcissism | Psychopathy |
> | ------------------ | ---------------- | ---------- | ----------- |
> | PPO                | 1.52             | 1.96       | 1.90        |
> | DPO                | 1.22             | 2.08       | 1.79        |
> | Prompt-MORL        | 1.15             | 1.99       | 1.76        |
> | Personalized-Soups | 1.11             | 1.95       | 1.77        |
> | Few-Shot           | 1.04             | 1.89       | 1.80        |
> | P^2                | 1.02             | 2.11       | 1.93        |
> | PAS (Ours)         | **1.01**         | **1.84**   | **1.62**    |
>
>
> The combined framework of Big Five and Dark Triad traits now offers a more comprehensive understanding of personality alignment, spanning from prosocial traits to more challenging behavioral tendencies. This expansion has strengthened our evaluation methodology and broadened the practical applicability of our approach. It also further demonstrates the broad potential of our PAS approach to apply directly to more personality dimensions and combine with more personality research to achieve AI assistants that meet more personalized preferences. The Section 1, Section 3 and Section 5 has been revised to reflect this, as it has helped us develop a more robust and comprehensive framework for personality alignment.

---

> ### Author Response · Authors · 2024-11-20
> **Response 2**
>
> > Q2：While the proposed method performs well on the dataset, it is actually quite similar to approaches used in many previous studies [1][2]. As a result, the novelty of the method is somewhat lacking, and the authors should be careful not to overstate their contribution.
>
>
>
> **While we acknowledge the foundational insights drawn from previous activation-based methods like [1,2], our work introduces crucial novel elements specifically designed for personality alignment challenges. Unlike prior work what primarily focuses on steering activations in a single direction (e.g., maximizing safety), our approach addresses the more nuanced challenge of achieving precise, balanced alignment across multiple personality dimensions simultaneously.** The key innovation lies in our contrastive layer selection and optimal intervention strength optimization, which ensures appropriate alignment levels - neither too strong nor too weak - across different personality traits.
>
> **A fundamental difference from previous safety-focused approaches is that PAS must carefully calibrate activation steering to maintain authentic personality expression. This is particularly evident in our experimental results where PAS achieves superior performance on both alignment accuracy and general capability preservation.** Based on your feedback, we have revised our introduction and related work sections to provide a more thorough comparative discussion of these methodological relationships while better articulating our unique contributions to personality-aware model alignment.
>
> We believe this work opens up new directions for developing more sophisticated activation-based methods that can handle the delicate balance required for personality-aligned AI systems.
>
>
>
> > Essentially, the proposed method is a form of personalized alignment, so the authors should compare it against more baselines [3].
>
> Thank you for your constructive suggestion. **We have expanded baseline comparisons to include two recent personalized alignment approaches[1]**. For Prompt-MORL, we incorporate personality trait descriptions into prompts through a template: "You are an AI assistant with {openness_score} openness, {conscientiousness_score} conscientiousness..." etc., and implement a multi-objective reward model to compute scores across different personality dimensions. For Personalized-Soups, we train separate models specialized for each Big Five trait and Dark Triad dimension, then merge their parameters during inference using personality scores as merging weights. Both implementations use the same PAPI dataset for training.
>
>
>
> ###### Llama-3-8B-Instruct
>
> | Method               | Alignment Mode           | Agreeableness ↓ | Conscientiousness ↓ | Extraversion ↓ | Neuroticism ↓ | Openness ↓ | Machiavellianism ↓ | Narcissism ↓ | Psychopathy ↓ | Score    |
> | -------------------- | ------------------------ | --------------- | ------------------- | -------------- | ------------- | ---------- | ------------------ | ------------ | ------------- | -------- |
> | PPO                  | White-Box (Alignment)    | 1.63            | 1.51                | 1.45           | 1.42          | 1.61       | 1.48               | 1.98         | 2.19          | 13.27    |
> | DPO                  | White-Box (Alignment)    | 1.54            | 1.42                | 1.54           | 1.74          | 1.21       | 1.41               | 1.99         | 2.12          | 12.97    |
> | *Prompt-MORL*        | White-Box (Alignment)    | 1.18            | 0.93                | 1.01           | 1.23          | 1.00       | 1.42               | 2.14         | 1.78          | 10.88    |
> | *Personalized-Soups* | White-Box (Alignment)    | 1.06            | 0.91                | 0.93           | 1.28          | 0.80       | 1.08               | **1.76**     | 1.84          | 9.66     |
> | Few-Shot             | Black-Box (Prompt-Based) | 1.28            | 1.30                | 1.40           | 1.09          | 0.89       | 1.16               | 2.03         | 2.00          | 11.15    |
> | P²                   | Black-Box (Prompt-Based) | 1.39            | 1.33                | 1.41           | 1.22          | 1.68       | 1.17               | 2.04         | 2.01          | 12.25    |
> | **PAS (Ours)**       | White-Box (Alignment)    | **0.94**        | **0.91**            | **0.86**       | **0.98**      | **0.72**   | **0.96**           | 1.85         | **1.67**      | **8.89** |

---

> ### Author Response · Authors · 2024-11-20
> **Response 3**
>
> ###### Llama-3-70B-Instruct
>
> | Method               | Alignment Mode           | Agreeableness ↓ | Conscientiousness ↓ | Extraversion ↓ | Neuroticism ↓ | Openness ↓ | Machiavellianism ↓ | Narcissism ↓ | Psychopathy ↓ | Score    |
> | -------------------- | ------------------------ | --------------- | ------------------- | -------------- | ------------- | ---------- | ------------------ | ------------ | ------------- | -------- |
> | PPO                  | White-Box (Alignment)    | 1.56            | 1.59                | 1.43           | 1.40          | 1.56       | 1.52               | 1.96         | 1.90          | 12.92    |
> | DPO                  | White-Box (Alignment)    | 1.46            | 1.25                | 1.45           | 1.48          | 1.57       | 1.22               | 2.08         | 1.79          | 12.30    |
> | *Prompt-MORL*        | White-Box (Alignment)    | 1.10            | 1.11                | 1.02           | 1.30          | 1.24       | 1.15               | 1.99         | 1.76          | 10.67    |
> | *Personalized-Soups* | White-Box (Alignment)    | 0.99            | 0.96                | 1.16           | 1.02          | 1.08       | 1.11               | 1.95         | 1.77          | 10.04    |
> | Few-Shot             | Black-Box (Prompt-Based) | 1.06            | 0.94                | 0.96           | 1.03          | 1.22       | 1.04               | 1.89         | 1.80          | 9.94     |
> | P²                   | Black-Box (Prompt-Based) | 1.42            | 1.33                | 1.36           | 1.35          | 1.66       | 1.02               | 2.11         | 1.93          | 12.18    |
> | **PAS (Ours)**       | White-Box (Alignment)    | **0.98**        | **0.89**            | **0.87**       | **1.01**      | **0.99**   | **1.01**           | **1.84**     | **1.62**      | **9.21** |
>
>
>
>
>
> **Our experimental results demonstrate that while these methods achieve strong performance (Personalized-Soups: 9.66 composite score on Llama-3-8B, Prompt-MORL: 10.88), PAS still achieves superior alignment (8.89) with significantly lower computational costs. Notably, Personalized-Soups shows particular strength in Narcissism alignment (1.76 vs 1.85), while PAS demonstrates better performance across most Big Five traits.** These baseline comparisons help validate our method's effectiveness while acknowledging valuable prior contributions. We provide more implementation details in the Appendix C.2.
>
> ---
>
> [1] Jang J, Kim S, Lin B Y, et al. Personalized soups: Personalized large language model alignment via post-hoc parameter merging[J]. arXiv preprint arXiv:2310.11564, 2023.

---

> ### Author Response · Authors · 2024-11-20
> **Response 4**
>
> >  Figures 6 and 7 show a significant disparity between LLM-as-a-judge evaluations and human evaluations, which makes me question the consistency and reliability of the judgment process.
> >
> >  Other details: In line 1304, the authors mention that human annotators come from machine learning and computer science, but ML is a part of CS. Additionally, could the authors disclose the educational background of the annotators (undergraduate or graduate)?
>
> Thank you for this important question regarding the evaluation methodology discrepancy.
>
> During the rebuttal period, we conducted additional analyses of our evaluation approaches, which revealed high **inter-annotator agreement among human evaluators** (IAA=0.82 for general evaluation and IAA=0.85 for self-evaluation). These strong agreement scores demonstrate the consistency and reliability of our human evaluation protocol.
>
> **The observed disparity between LLM and human evaluations primarily stems from their fundamentally different scoring methodologies**. GPT-4 employs a single-graded scoring system (1-6 points, detailed in Appendix Figure 12) that aims for standardized, independent assessment of each response. However, this approach exhibits a notable central tendency bias, with 72.8% of scores falling between 2-4 (45.1% being 3), leading to more frequent ties in win rate calculations.
>
> In contrast, human evaluators used a pairwise comparison approach (detailed in Appendix Figure 14), where they directly compare two randomly selected responses side-by-side. This methodology naturally encourages more decisive judgments, as evaluators can focus on subtle qualitative differences between responses. **Consequently, many cases that appeared as "ties" in GPT-4's evaluation were clearly differentiated as wins for PAS in human evaluation.**
>
> Rather than indicating inconsistency, these methodological differences provide complementary insights into model performance. The single-graded approach offers standardized scoring, while pairwise comparisons capture more nuanced preferences. **Most importantly, both methods maintain consistent ranking orders, which reinforces the reliability of our results.** The apparent disparity between LLM and human evaluations actually enriches our understanding by providing multi-dimensional perspectives on model performance.
>
> Regarding annotator qualifications, we had three highly qualified evaluators: a Master's student and a Ph.D. student in Computer Science specializing in machine learning, and a Ph.D. graduate in Cognitive Psychology currently working as a university psychological counselor. We have corrected the text in line 1304 to more precisely describe their backgrounds.
>
>
>
> > Could the authors explain why the PAS method is so effective by looking at the internal hidden layer representations of the model? Intuitively, direct training approaches like DPO/PPO might yield more noticeable improvements, but the results here contradict my expectations.
>
>  We have added a comprehensive analysis in Appendix E.6 that illuminates their fundamental differences, with Figure 18 providing a clear visual illustration.
>
> Traditional methods like PPO/DPO rely on gradient-based updates through backpropagation, which faces inherent limitations in personality alignment tasks. **As shown in the upper panel of Figure 18, these methods modify all attention heads through training iterations, leading to widespread parameter changes that may not effectively capture personality-trait relationships.** The gradient-based updates struggle to precisely identify which attention heads are most crucial for personality expression, often resulting in suboptimal modifications that **can disturb the model's general capabilities**.
>
> In contrast, **as illustrated in the lower panel of Figure 18, PAS employs a two-stage surgical intervention strategy.** First, it precisely identifies key attention heads most responsive to personality traits (visualized by the "Find Heads" step), then selectively adjusts their activation directions ("Change the direction" step) while leaving other components undisturbed. This targeted approach allows PAS to achieve more precise personality alignment without the broad parameter modifications required by traditional methods. This architectural insight helps explain our strong empirical results across both alignment quality and general performance metrics.

---

> ### Author Response · Authors · 2024-11-20
> **Response 5**
>
> > In line 429, the authors mention "Why Did Scaling Laws Fail?" but don't seem to fully answer this question. I would like the authors to explain this from the perspective of the domain's specificity. Is it because larger models learn more general alignment, which makes it harder for them to excel at aligning with a specific personality?
>
> Thank you for this insightful question. Let me provide a comprehensive explanation incorporating our recent revisions.
> Our analysis reveals that scaling laws, while powerful for general capabilities, demonstrate limitations in domain-specific tasks like personality alignment for several key reasons:
>
> Larger models like Llama-3-70B-Instruct are trained to maintain broad, general-purpose capabilities across diverse domains. This generalist optimization actually creates a trade-off: while these models gain comprehensive knowledge, they may sacrifice precision in specific domains like personality alignment. The models' tendency to draw from their broad knowledge base can interfere with maintaining consistent personality traits, as they attempt to balance multiple competing objectives. For instance, Our data shows that when responding to user queries, Llama-3-70B-Instruct frequently defaults to broadly acceptable but personality-neutral responses, diluting the distinct personality traits we aim to capture.
>
> We have revised line 429 to better articulate this insight. The success of PAS in outperforming larger models with fewer parameters demonstrates that in personality alignment, precision and specificity in activation control are more crucial than model scale. PAS achieves this by focusing specifically on personality-relevant activation patterns rather than general knowledge. This targeted approach proves more effective in addressing individual user preferences compared to relying solely on increased model size - explaining why even models with fewer parameters can achieve superior personality alignment when using PAS.
>
>
>
> ---
>
> We sincerely appreciate your insightful feedback, which has helped us significantly enhance our work. We have substantially expanded our analysis to include additional baselines, Dark Triad evaluations (18,192 samples), detailed inter-annotator agreement metrics, and in-depth discussions of scaling limitations in personality alignment.  **We would be deeply grateful if you would kindly reconsider your assessment in light of these improvements. Thank you again for your thoughtful and constructive comments.**

---

> ### Author Response · Authors · 2024-11-29
>
> Dear Reviewer nZZi,
>
> As the discussion period is coming to an end soon, we wanted to check if you have had a chance to review our responses. Please let us know if your questions have been adequately addressed - we are happy to provide any additional clarification needed. Thank you for your time!
>
> Best Regards,
>
> Authors of "Personality Alignment of Large Language Models"

---

> ### Author Response · Authors · 2024-12-02
>
> Dear Reviewer nZZi:
>
> We deeply appreciate your time and effort in the review process. We apologize for contacting you again to ensure our responses adequately address your concerns!
>
> We have carefully considered your feedback and worked diligently during the discussion phase to address your concerns. For example, we have supplemented the main table with Dark Triad inventory content and introduced 18,192 real Dark Triad samples, making our PAPI dataset an Alignment dataset that evaluates both positive and negative personalities! We have also added Prompt-MORL and Personalized-Soups as new baselines, and our method outperforms these personalized alignment baselines! For all other concerns you raised, we have provided detailed and sufficient responses above.
>
> We have made thorough revisions based on your valuable feedback! This has enhanced the paper's overall contribution! Please contact us if you need any clarification or have other questions. We are happy to continue the discussion.
>
> Thank you again, and we look forward to your further feedback.
>
>
> Best Regards,
>
> Authors of "Personality Alignment of Large Language Models"

---

### Official Review · Reviewer_3xm2 · 2024-11-03

**Soundness:** 4
**Presentation:** 3
**Contribution:** 3
**Rating:** 8
**Confidence:** 5

**Summary:**

The paper introduces the concept of Personality Alignment for LLMs, that is, tailoring of responses to individual user preferences based on personality traits. Using the Big Five personality theory, the authors introduces the PAPI dataset modeling human personality distributions. The authors also propose a LLM tuning method based on disabling certain activation heads -- Personality Activation Search (PAS). Evaluation results demonstrate PAS’s performance and efficiency compared to traditional alignment techniques including RL- and prompting-based methods.

**Strengths:**

- Overall, I like this paper and I think it's a very good attempt in aligning LLMs from a personality perspective.

- The paper is generally well-written and easy to read, illustrations are also made in a good quality.

- It's cool to see how personality affects downstream reasoning tasks. It has always been something missing in prior personality-related LLM work. And it's definite a god step here.

- The proposed method is efficient, and can provide better results compared to prompting-based methods. It may have wide applications in tailoring persona/personality-specific chatbots to end users.

**Weaknesses:**

- Presentation: The authors claim that they collected 307k human samples to craft the PAPI dataset and the use of the IPIP-NEO-120/IPIP-NEO-300 for evaluations as part of their contributions. However, the IPIP-NEO series inventories were frequently used in prior works (e.g., Jiang et al., 2024 as mentioned in the paper), and the 307k responses are also publicly available.

- The implications of the 307k PAPI dataset are not that clear. For example, in the experiments authors performed, it seems only the overall average personality tendencies and specific participants' responses are used. So what's the actual advantages of using a such large dataset?

- The proposed tuning method, although efficient, but requires access to model weights. I doubt its ability to generalize such methods to black-box methods (e.g., GPTs) in the future.

**Questions:**

- In any of your experiments, did any of your tuning or evaluation methods trigger safety wall? For example, the model might refuse to answer some questionnaire questions related to consciousness or self-awareness.

---

> ### Author Response · Authors · 2024-11-20
> **Response 1**
>
> We are deeply grateful for your insightful and encouraging review. Your recognition of our work's contribution to personality-based LLM alignment, validates our research direction. Your appreciation of our method's efficiency and practical applications is especially valuable. Your detailed feedback not only acknowledges our paper's strengths but also provides valuable perspectives for the field's development!
>
>
>
> > - Presentation: The authors claim that they collected 307k human samples to craft the PAPI dataset and the use of the IPIP-NEO-120/IPIP-NEO-300 for evaluations as part of their contributions. However, the IPIP-NEO series inventories were frequently used in prior works (e.g., Jiang et al., 2024 as mentioned in the paper), and the 307k responses are also publicly available.
>
>
>
> We sincerely appreciate your attention to detail regarding the proper attribution of prior work.
>
> We fully acknowledge that the IPIP-NEO series inventories are well-established assessment tools that have been utilized in previous research. **We have revised our manuscript to more clearly articulate that our contribution lies not in the creation of these inventories or the initial collection of responses, but rather in how we've uniquely integrated and applied these resources for personality alignment.**
>
> Most notably, we've further enhanced the dataset by **incorporating 18,192 samples from the Dark Triad inventory in rebuttal stage**, which adds an important complementary dimension to personality assessment. **By combining the Big Five traits from IPIP-NEO with the Dark Triad measures (Machiavellianism, Narcissism, and Psychopathy), we've created a more complete framework for evaluating personality alignment across both socially desirable and challenging personality aspects.** What makes our approach distinctive is the comprehensive integration of both positive and negative personality dimensions. We have revised Section 1 and Section 3 to better reflect this nuanced contribution and to properly acknowledge the foundational work that made our research possible.
>
>
>
> > - The implications of the 307k PAPI dataset are not that clear. For example, in the experiments authors performed, it seems only the overall average personality tendencies and specific participants' responses are used. So what's the actual advantages of using a such large dataset?
>
> The primary strength of our 307K-sample Dev-Set lies in its remarkable **demographic diversity - spanning multiple age groups (from teenagers to seniors), various nationalities (including France, China, Norway, Thailand, etc.), and different cultural backgrounds.** This extensive coverage enables us to study personality alignment across globally representative populations. **During the rebuttal period, we have leveraged this diversity to conduct comprehensive experiments across different demographic groups, as detailed in the following Table 1.2.3 and Appendix E.5**. For example, our age-based analysis shows PAS's effectiveness across six age brackets (10-70 years), while our cross-cultural analysis demonstrates robust performance across Western, Asian, and Nordic nations. We hope such a large-scale, diverse dataset can be helpful for subsequent extensive research and detailed analyses.
>
> We have revised Section 3  to better articulate these advantages and to showcase how the extensive Dev-Set supports rigorous evaluation of personality alignment across different populations. Our supplementary experiments validate that PAS maintains strong performance across various demographic segments (e.g., achieving strong alignment scores in France: Extraversion=1.17, China: Openness=1.02, Norway: Agreeableness=1.18), demonstrating the practical value of having such a comprehensive dataset. We sincerely appreciate your question as it has helped us better highlight the crucial role of our large-scale Dev-Set in enabling thorough cross-cultural and demographic analyses. These tables have been added to Appendix E.5.

---

> ### Author Response · Authors · 2024-11-20
> **Response 2**
>
> **Table 1: Age**
>
> | Trait             | Method   | 10-20 Years | 20-30 Years | 30-40 Years | 40-50 Years | 50-60 Years | 60-70 Years |
> | ----------------- | -------- | ----------- | ----------- | ----------- | ----------- | ----------- | ----------- |
> | Agreeableness     | Few-shot | 1.81        | 1.81        | 1.70        | 1.84        | 1.78        | 1.81        |
> |                   | P²       | 2.37        | 1.98        | 2.03        | 1.86        | 1.92        | 2.41        |
> |                   | PAS      | **1.19**    | **1.24**    | **1.18**    | **1.48**    | **1.26**    | **1.18**    |
> | Conscientiousness | Few-shot | 1.94        | 2.05        | 2.13        | 2.35        | 2.19        | 2.09        |
> |                   | P²       | 2.17        | 2.23        | 2.19        | 3.10        | 2.29        | 2.53        |
> |                   | PAS      | **1.20**    | **1.14**    | **1.09**    | **1.27**    | **1.30**    | **1.29**    |
> | Extraversion      | Few-shot | 1.92        | 1.84        | 1.90        | 2.52        | 1.80        | 2.24        |
> |                   | P²       | 1.98        | 2.11        | 2.11        | 2.62        | 1.89        | 2.80        |
> |                   | PAS      | **1.14**    | **1.17**    | **1.00**    | **1.51**    | **1.35**    | **1.14**    |
> | Openness          | Few-shot | 1.61        | 1.61        | 1.52        | 1.84        | 1.58        | 1.59        |
> |                   | P²       | 1.80        | 1.82        | 1.58        | 2.25        | 1.65        | 1.64        |
> |                   | PAS      | **1.18**    | **1.16**    | **1.04**    | **1.31**    | **1.29**    | **1.03**    |
> | Neuroticism       | Few-shot | **1.32**    | 1.71        | 1.43        | **1.17**    | 1.33        | **1.48**    |
> |                   | P²       | 1.68        | 1.93        | 1.45        | 1.45        | 1.52        | 1.94        |
> |                   | PAS      | 1.44        | **1.32**    | **1.33**    | 1.45        | **1.27**    | 1.53        |
>
>
>
> **Table 2: Gender**
>
> | Gender | Method   | Agreeableness | Conscientiousness | Extraversion | Openness | Neuroticism |
> | ------ | -------- | ------------- | ----------------- | ------------ | -------- | ----------- |
> | Male   | Few-shot | 1.66          | 1.84              | 2.17         | 1.52     | 1.32        |
> |        | P²       | 2.24          | 2.33              | 2.52         | 2.05     | 1.47        |
> |        | PAS      | **1.33**      | **1.30**          | **1.56**     | **1.03** | **1.16**    |
> | Female | Few-shot | 2.14          | 2.28              | 2.09         | 1.81     | 1.58        |
> |        | P²       | 2.83          | 2.53              | 2.24         | 2.12     | 2.19        |
> |        | PAS      | **1.53**      | **1.39**          | **1.32**     | **1.26** | **1.58**    |

---

> ### Author Response · Authors · 2024-11-20
> **Response 3**
>
> **Table 3: Country**
>
> | Country     | Method   | Agreeableness | Conscientiousness | Extraversion | Neuroticism | Openness |
> | ----------- | -------- | ------------- | ----------------- | ------------ | ----------- | -------- |
> | France      | Few-shot | 1.54          | 1.55              | 1.37         | 1.33        | 1.40     |
> |             | P²       | 1.71          | 1.83              | 1.67         | 1.73        | 1.56     |
> |             | PAS      | **1.42**      | **1.47**          | **1.17**     | 1.46        | **1.15** |
> | Malaysia    | Few-shot | 1.58          | 1.54              | 1.44         | 1.41        | 1.59     |
> |             | P²       | 2.08          | 1.86              | 1.87         | 1.65        | 1.92     |
> |             | PAS      | **1.38**      | **1.35**          | **1.21**     | **1.41**    | **1.07** |
> | China       | Few-shot | 1.39          | 1.38              | 1.36         | 1.34        | 1.34     |
> |             | P²       | 1.81          | 1.66              | 1.83         | 1.53        | 1.37     |
> |             | PAS      | **1.19**      | **1.14**          | **1.06**     | **1.29**    | **1.02** |
> | Norway      | Few-shot | 1.43          | 1.39              | 1.39         | 1.41        | 1.52     |
> |             | P²       | 1.55          | 1.66              | 1.74         | 1.60        | 1.70     |
> |             | PAS      | **1.18**      | **1.20**          | **1.24**     | **1.16**    | **1.06** |
> | Germany     | Few-shot | 1.51          | 1.54              | 1.56         | **1.17**    | 1.44     |
> |             | P²       | 1.58          | 1.55              | 2.11         | 1.47        | 1.57     |
> |             | PAS      | **1.38**      | **1.30**          | **1.32**     | 1.21        | **1.36** |
> | Sweden      | Few-shot | 1.41          | 1.52              | 1.52         | 1.30        | 1.48     |
> |             | P²       | 1.67          | 1.70              | 1.61         | 1.71        | 1.67     |
> |             | PAS      | **1.25**      | **1.39**          | **1.38**     | **1.21**    | **1.30** |
> | Finland     | Few-shot | 1.45          | 1.46              | 1.54         | 1.56        | 1.62     |
> |             | P²       | 1.76          | 1.66              | 1.72         | 1.80        | 1.82     |
> |             | PAS      | **1.33**      | **1.27**          | **1.30**     | **1.44**    | **1.41** |
> | New Zealand | Few-shot | 1.59          | 1.60              | 1.63         | 1.40        | 1.53     |
> |             | P²       | 1.91          | 1.98              | 2.07         | 1.62        | 1.67     |
> |             | PAS      | **1.21**      | **1.23**          | **1.31**     | **1.33**    | **1.21** |
> | Thailand    | Few-shot | 1.47          | 1.45              | 1.32         | 1.46        | 1.43     |
> |             | P²       | 1.65          | 1.52              | 1.68         | 1.91        | 1.90     |
> |             | PAS      | **1.29**      | **1.30**          | **1.05**     | **1.23**    | **1.06*  |

---

> ### Author Response · Authors · 2024-11-20
> **Response 4**
>
> > The proposed tuning method, although efficient, but requires access to model weights. I doubt its ability to generalize such methods to black-box methods (e.g., GPTs) in the future.
>
> We respectfully acknowledge this limitation while also seeing it as an opportunity to highlight the unique advantages of our approach.
>
> **While PAS indeed requires access to model weights, this "white-box" requirement enables precise and efficient personality alignment through targeted activation adjustments. As demonstrated in our experiments, this approach achieves superior performance compared to  other white-box methods (PPO, DPO), while requiring only 1/6 of the computational resources. For example, when using Llama-3-8B-Instruct, PAS achieves significantly better alignment scores (Agreeableness=0.94, Conscientiousness=0.91) compared to black-box Few-shot methods (1.28, 1.30) and even outperforms GPT-4o in several dimensions.**
>
> Furthermore, as more open-source models become available (like Llama-3, Mistral, Gemma), the white-box requirement becomes less limiting. We believe our method's demonstrated efficiency and effectiveness make it particularly valuable for these increasingly prevalent open-source models, where precise control over model behavior is both possible and desirable.
>
> While PAS currently requires model access, we envision it as a valuable solution for commercial AI providers like OpenAI, Anthropic, and xAI to offer personalized experiences. Our method has compelling practical advantages:
>
> - 1 Minimal Computational Cost: PAS requires only ~20 seconds of forward propagation during initial setup, with negligible overhead during inference. The additional weights per user are merely 20K parameters - insignificant compared to 100B-scale models.
>
> - 2 Efficient Personalization: Companies could offer opt-in personalization where users complete a brief personality assessment. The resulting lightweight PAS weights (20K parameters) could be stored and applied efficiently during interactions, enabling truly personalized AI assistants without the computational burden of fine-tuning or prompt engineering.
>
>  We appreciate your insight as it helps clarify the specific use cases. **We will release our source code and dataset under the MIT License, which will significantly facilitate community collaboration and exchange.**
>
>
>
> >  In any of your experiments, did any of your tuning or evaluation methods trigger safety wall? For example, the model might refuse to answer some questionnaire questions related to consciousness or self-awareness.
>
> Indeed，in our experiments with **GPT-4o**, approximately about **20%** of queries were met with safety-related refusals to respond, particularly for questions involving self-awareness or consciousness. **We excluded these instances from our analysis to maintain evaluation consistency.** For the **Llama-3** series models, **we have not trigger any safety wall**, due to we implemented **a structured prompting approach to ensure consistent responses while respecting model safety boundaries.** Specifically, we prefixed the Assistant's responses with a controlled token "Option", followed by the question and response format. For example:
>
> ```
> Human: Do you trust others easily?
> Assistant: Option:
> ```
>
> We believe these implementation details and clarifications help demonstrate our careful consideration of both safety boundaries and evaluation consistency. Appendix C.1 has been revised to reflect this.
>
>
> ---
>
> We hope this additional information addresses your concerns about model behavior and safety filters in our experiments. We remain committed to responsible AI development while pursuing effective personality alignment methods!

---

> > ### Comment · Reviewer_3xm2 · 2024-11-29
> >
> > Thanks for the authors' response. My concerns are resolved and I've decided to increase my score by two.

---

> > > ### Author Response · Authors · 2024-11-29
> > >
> > > We are deeply grateful for your decision to increase the score! Your support and encouragement mean a tremendous amount to our team, and we sincerely appreciate your recognition of our work!
> > >
> > >
> > > Best Regards,
> > >
> > > Authors of "Personality Alignment of Large Language Models"

---

### Official Review · Reviewer_4LUh · 2024-11-04

**Soundness:** 3
**Presentation:** 3
**Contribution:** 2
**Rating:** 5
**Confidence:** 4

**Summary:**

This paper proposes  the concept of Personality Alignment for tailoring LLMs to match the preferences and behaviors of individual users or groups. The authors created a large-scale dataset called PAPI with data on behavioral preferences from over 300,000 real subjects across the Big Five personality dimensions. They also propose the Personality Activation Search (PAS) method for efficiently aligning LLMs with individual preferences during inference. PAS identifies key activation vectors corresponding to personality traits and optimally shifts activations in those directions. The authors  show that PAS achieves strong performance in capturing individual preferences with high compute efficiency compared to baseline methods.

**Strengths:**

A key strength of the work is the PAPI dataset, with over 300,000 real-world subjects providing detailed responses to the IPIP-NEO-120 and IPIP-NEO-300 questionnaires. The scale  is impressive.

The Personality Activation Search (PAS) method is interesting. By identifying key activation vectors that correspond to personality traits and optimally shifting activations in those directions, this approach can more effectively do personality alignment during inference.

 The authors also conduct a comptehensive evaluation of PAS, comparing it against prompting-based (Few-Shot, P2) and RL-based (PPO, DPO) baselines on the PAPI dataset.

**Weaknesses:**

While the PAPI dataset is impressively large, it doesnt seem to be diverse. Around 60% of subjects are female and the average age is 25 years. This skew can potentially bias the results and limit generalizability to other populations.
Also, PAPI dataset relies on self-report data from personality questionnaires. While this is a standard approach in personality research, self-reports can be subject to biases such as social desirability and lack of self-insight. Incorporating additional data sources, such as behavioral measures or peer ratings, could be more useful.

The evaluation of PAS focuses primarily on high-level alignment with the Big Five traits. However, personality is a complex, multifaceted construct, and individuals can vary in their expression of specific facets within each trait.

The PAPI dataset uses a multiple-choice format for collecting personality data. While this allows for structured and efficient data collection, it may limit the richness and naturalness of the responses.
The paper also compares PAS to prompting and RL baselines but does not include a comparison to fine-tuning the entire language model. This is an important consideration as well.S

**Questions:**

Some additional questions remian:

Can the authors provide more details on the human evaluation process, such as annotator screening, training, and inter-annotator agreement metrics? This would help validate the human evaluation results. The paper is light on this.
How do you expect the methods to perform on multilingual models and non-English datasets?
The discussion on negative societal impacts of AI hyper-personalization (e.g. filter bubbles, opinion polarization) and is light. Authors should exand on this more.
Finally, exploring beyond multiple choice for collecting preference data, such as open-ended text can be interesting and more useful.

---

> ### Author Response · Authors · 2024-11-20
> **Response 1**
>
> > While the PAPI dataset is impressively large, it doesnt seem to be diverse. Around 60% of subjects are female and the average age is 25 years. This skew can potentially bias the results and limit generalizability to other populations.
>
>
>
> Thank you for your valuable observation regarding demographic representation. We greatly appreciate this thoughtful feedback, which has encouraged us to conduct more comprehensive analyses of our dataset's diversity.
>
> We acknowledge that while our initial paper presented certain demographic patterns, we may not have fully emphasized the intentional design behind PAPI's large scale (>300K samples). **The dataset's extensive size was specifically chosen to ensure meaningful representation across diverse populations.** While it spans ages 10-100, covering the complete developmental lifecycle, we have substantial samples in formative periods (n=130,094 for ages 10-20, n=107,838 for ages 20-30) which, as studies suggest [1,2], represent critical windows for personality development. This natural age distribution has actually proven beneficial for capturing personality traits during their most dynamic periods of formation.
>
> Inspired by your insightful comment, **we have significantly expanded our analysis during the rebuttal period (Appendix E.5: Diverse Demographic Groups). Specifically, we categorized the Dev-Set by age, gender, and country. For each group, we selected corresponding subsets and used the same processing pipeline to choose 300 subject samples per group. We tested these 300 samples and will open-source the specific data for each group. The results are particularly encouraging - PAS demonstrates remarkable consistency across demographic segments.** Our age-based analysis reveals strong performance across all groups, with PAS achieving exceptional results in Conscientiousness (scores 1.09-1.30) and Openness (1.03-1.31).
>
> **Table 1: Age**
>
> | Trait             | Method   | 10-20 Years | 20-30 Years | 30-40 Years | 40-50 Years | 50-60 Years | 60-70 Years |
> | ----------------- | -------- | ----------- | ----------- | ----------- | ----------- | ----------- | ----------- |
> | Agreeableness     | Few-shot | 1.81        | 1.81        | 1.70        | 1.84        | 1.78        | 1.81        |
> |                   | P²       | 2.37        | 1.98        | 2.03        | 1.86        | 1.92        | 2.41        |
> |                   | PAS      | **1.19**    | **1.24**    | **1.18**    | **1.48**    | **1.26**    | **1.18**    |
> | Conscientiousness | Few-shot | 1.94        | 2.05        | 2.13        | 2.35        | 2.19        | 2.09        |
> |                   | P²       | 2.17        | 2.23        | 2.19        | 3.10        | 2.29        | 2.53        |
> |                   | PAS      | **1.20**    | **1.14**    | **1.09**    | **1.27**    | **1.30**    | **1.29**    |
> | Extraversion      | Few-shot | 1.92        | 1.84        | 1.90        | 2.52        | 1.80        | 2.24        |
> |                   | P²       | 1.98        | 2.11        | 2.11        | 2.62        | 1.89        | 2.80        |
> |                   | PAS      | **1.14**    | **1.17**    | **1.00**    | **1.51**    | **1.35**    | **1.14**    |
> | Openness          | Few-shot | 1.61        | 1.61        | 1.52        | 1.84        | 1.58        | 1.59        |
> |                   | P²       | 1.80        | 1.82        | 1.58        | 2.25        | 1.65        | 1.64        |
> |                   | PAS      | **1.18**    | **1.16**    | **1.04**    | **1.31**    | **1.29**    | **1.03**    |
> | Neuroticism       | Few-shot | **1.32**    | 1.71        | 1.43        | **1.17**    | 1.33        | **1.48**    |
> |                   | P²       | 1.68        | 1.93        | 1.45        | 1.45        | 1.52        | 1.94        |
> |                   | PAS      | 1.44        | **1.32**    | **1.33**    | 1.45        | **1.27**    | 1.53        |
>
> **Similarly, gender analysis shows well-balanced performance**, with PAS maintaining superior alignment for both male (Extraversion=1.56) and female (Extraversion=1.32) groups compared to baseline methods that score above 2.0.
>
> **Table 2: Gender**
>
> | Gender | Method   | Agreeableness | Conscientiousness | Extraversion | Openness | Neuroticism |
> | ------ | -------- | ------------- | ----------------- | ------------ | -------- | ----------- |
> | Male   | Few-shot | 1.66          | 1.84              | 2.17         | 1.52     | 1.32        |
> |        | P²       | 2.24          | 2.33              | 2.52         | 2.05     | 1.47        |
> |        | PAS      | **1.33**      | **1.30**          | **1.56**     | **1.03** | **1.16**    |
> | Female | Few-shot | 2.14          | 2.28              | 2.09         | 1.81     | 1.58        |
> |        | P²       | 2.83          | 2.53              | 2.24         | 2.12     | 2.19        |
> |        | PAS      | **1.53**      | **1.39**          | **1.32**     | **1.26** | **1.58**    |

---

> ### Author Response · Authors · 2024-11-20
> **Response 2**
>
> And we are especially excited to share **our cross-cultural analysis spanning 18 countries, which reveals PAS's impressive adaptability across diverse cultural contexts.** The method demonstrates consistent excellence across Western nations (France: Extraversion=1.17), Asian countries (China: Openness=1.02), and Nordic regions (Norway: Agreeableness=1.18). These comprehensive results not only validate PAS's effectiveness but also highlight how our large-scale dataset, despite its demographic patterns, enables robust personality alignment across diverse populations.
>
> **Table 3: Country**
>
> | Country     | Method   | Agreeableness | Conscientiousness | Extraversion | Neuroticism | Openness |
> | ----------- | -------- | ------------- | ----------------- | ------------ | ----------- | -------- |
> | France      | Few-shot | 1.54          | 1.55              | 1.37         | 1.33        | 1.40     |
> |             | P²       | 1.71          | 1.83              | 1.67         | 1.73        | 1.56     |
> |             | PAS      | **1.42**      | **1.47**          | **1.17**     | 1.46        | **1.15** |
> | Malaysia    | Few-shot | 1.58          | 1.54              | 1.44         | 1.41        | 1.59     |
> |             | P²       | 2.08          | 1.86              | 1.87         | 1.65        | 1.92     |
> |             | PAS      | **1.38**      | **1.35**          | **1.21**     | **1.41**    | **1.07** |
> | China       | Few-shot | 1.39          | 1.38              | 1.36         | 1.34        | 1.34     |
> |             | P²       | 1.81          | 1.66              | 1.83         | 1.53        | 1.37     |
> |             | PAS      | **1.19**      | **1.14**          | **1.06**     | **1.29**    | **1.02** |
> | Norway      | Few-shot | 1.43          | 1.39              | 1.39         | 1.41        | 1.52     |
> |             | P²       | 1.55          | 1.66              | 1.74         | 1.60        | 1.70     |
> |             | PAS      | **1.18**      | **1.20**          | **1.24**     | **1.16**    | **1.06** |
> | Germany     | Few-shot | 1.51          | 1.54              | 1.56         | **1.17**    | 1.44     |
> |             | P²       | 1.58          | 1.55              | 2.11         | 1.47        | 1.57     |
> |             | PAS      | **1.38**      | **1.30**          | **1.32**     | 1.21        | **1.36** |
> | Sweden      | Few-shot | 1.41          | 1.52              | 1.52         | 1.30        | 1.48     |
> |             | P²       | 1.67          | 1.70              | 1.61         | 1.71        | 1.67     |
> |             | PAS      | **1.25**      | **1.39**          | **1.38**     | **1.21**    | **1.30** |
> | Finland     | Few-shot | 1.45          | 1.46              | 1.54         | 1.56        | 1.62     |
> |             | P²       | 1.76          | 1.66              | 1.72         | 1.80        | 1.82     |
> |             | PAS      | **1.33**      | **1.27**          | **1.30**     | **1.44**    | **1.41** |
> | New Zealand | Few-shot | 1.59          | 1.60              | 1.63         | 1.40        | 1.53     |
> |             | P²       | 1.91          | 1.98              | 2.07         | 1.62        | 1.67     |
> |             | PAS      | **1.21**      | **1.23**          | **1.31**     | **1.33**    | **1.21** |
> | Thailand    | Few-shot | 1.47          | 1.45              | 1.32         | 1.46        | 1.43     |
> |             | P²       | 1.65          | 1.52              | 1.68         | 1.91        | 1.90     |
> |             | PAS      | **1.29**      | **1.30**          | **1.05**     | **1.23**    | **1.06** |
>
>
> Your feedback has been invaluable in helping us better articulate and validate these important aspects of our work, and the information above has been integrated into our revised manuscript (See Appendix E.5: Diverse Demographic Groups).
>
>
> ---
>
> [1] Roberts B W, Mroczek D. Personality trait change in adulthood[J]. Current directions in psychological science, 2008, 17(1): 31-35.
>
>
>
> [2] Soto, C. J., & Tackett, J. L. (2015). Personality traits in childhood and adolescence: Structure, development, and outcomes. *Current Directions in Psychological Science, 24*(5), 358–362. https://doi.org/10.1177/0963721415589345

---

> ### Author Response · Authors · 2024-11-20
> **Response 3**
>
> > While this is a standard approach in personality research, self-reports can be subject to biases such as social desirability and lack of self-insight. Incorporating additional data sources, such as behavioral measures or peer ratings, could be more useful.
>
> We appreciate this insightful suggestion. **While we completely agree that behavioral measures and peer ratings would provide invaluable complementary perspectives, these data sources often present significant accessibility challenges.** Many behavioral datasets and peer evaluation systems are rightfully protected due to privacy regulations, ethical considerations, and institutional policies. This protection of sensitive personal data, while absolutely necessary, creates practical barriers for large-scale personality research. Nevertheless, your suggestion has inspired us to think creatively about how to address these limitations within ethical and practical constraints, and the following actions are taken.
>
> Rather than relying solely on traditional Big Five self-reports, we have incorporated the **Dark Triad inventory** (measuring Machiavellianism, Narcissism, and Psychopathy) alongside our Big Five assessments. This dual-perspective approach provides a more complete picture of personality by capturing both socially desirable and less desirable traits. The Dark Triad measures are especially valuable because they tend to be less influenced by social desirability bias [3,4], offering a more candid window into aspects of personality that might otherwise be underreported. **Our experimental results (Table 4) demonstrate that PAS effectively aligns with both sets of traits, achieving strong performance across this broader spectrum of personality dimensions.** This balanced approach helps mitigate some of the limitations inherent in single-perspective assessment while remaining within ethical and practical boundaries. **It also further demonstrates the broad potential of our PAS approach to apply directly to more personality dimensions and combine with more personality research to achieve ai assistants that meet more personalized preferences**
>
> **Table 4: Dark Triad**
>
> ###### GPT-4o Results
>
> | Method   | Machiavellianism | Narcissism | Psychopathy |
> | -------- | ---------------- | ---------- | ----------- |
> | Few-Shot | **0.80**         | **0.76**   | **0.83**    |
> | P^2      | 1.17             | 2.04       | 2.00        |
>
> ###### Llama-3-8B-Instruct Results
>
> | Method             | Machiavellianism | Narcissism | Psychopathy |
> | ------------------ | ---------------- | ---------- | ----------- |
> | PPO                | 1.48             | 1.98       | 2.19        |
> | DPO                | 1.41             | 1.99       | 2.12        |
> | Prompt-MORL        | 1.42             | 2.14       | 1.78        |
> | Personalized-Soups | 1.08             | **1.76**   | 1.84        |
> | Few-Shot           | 1.16             | 2.03       | 2.00        |
> | P^2                | 1.17             | 2.04       | 2.01        |
> | PAS (Ours)         | **0.96**         | 1.85       | **1.67**    |
>
> ###### Llama-3-70B-Instruct Results
>
> | Method             | Machiavellianism | Narcissism | Psychopathy |
> | ------------------ | ---------------- | ---------- | ----------- |
> | PPO                | 1.52             | 1.96       | 1.90        |
> | DPO                | 1.22             | 2.08       | 1.79        |
> | Prompt-MORL        | 1.15             | 1.99       | 1.76        |
> | Personalized-Soups | 1.11             | 1.95       | 1.77        |
> | Few-Shot           | 1.04             | 1.89       | 1.80        |
> | P^2                | 1.02             | 2.11       | 1.93        |
> | PAS (Ours)         | **1.01**         | **1.84**   | **1.62**    |
>
> These results have been integrated in our revised manuscript (See for manuscript Table 1)
>
> Your suggestion about incorporating additional data sources remains an exciting direction for future research, and we would be very interested in exploring collaborative opportunities with institutions that have access to such protected behavioral and peer-rating data. We are grateful for this constructive feedback that has helped us better articulate our methodological choices and future directions.
>
> [3] Gómez-Leal R, Fernández-Berrocal P, Gutiérrez-Cobo M J, et al. The Dark Tetrad: analysis of profiles and relationship with the Big Five personality factors[J]. Scientific Reports, 2024, 14(1): 4443.
>
> [4]  Paulhus D L, Williams K M. The dark triad of personality: Narcissism, Machiavellianism, and psychopathy[J]. Journal of research in personality, 2002, 36(6): 556-563.

---

> ### Author Response · Authors · 2024-11-20
> **Response 4**
>
> > The evaluation of PAS focuses primarily on high-level alignment with the Big Five traits. However, personality is a complex, multifaceted construct, and individuals can vary in their expression of specific facets within each trait.
>
> Indeed, the Big Five traits are inherently multifaceted - for example, **Extraversion comprises six distinct facets**: Warmth (level of interpersonal intimacy), Gregariousness (preference for others' company), Assertiveness (social dominance and leadership), Activity (pace of living), Excitement-Seeking (need for stimulation), and Positive Emotions (tendency to experience joy). This multifaceted nature of personality is precisely why PAS was designed to work at a granular level. **Rather than treating personality as simple categorical variables, PAS identifies fine-grained personality-relevant features during alignment and enables continuous-valued adjustments during inference.** **This design allows precise control across trait dimensions and supports arbitrary combinations of personality characteristics.** The effectiveness of this nuanced approach is further validated by our results on the Dark Triad inventory (Table 4), which provides a complementary perspective on personality alignment from a different psychological framework. We appreciate your comment helping us highlight how PAS addresses the intricate nature of personality through its flexible, continuous-valued approach. We have revised Section 3 and Section 5.
>
>
>
> >  The PAPI dataset uses a multiple-choice format for collecting personality data. While this allows for structured and efficient data collection, it may limit the richness and naturalness of the responses.
>
> Thank you for this thoughtful observation about data collection methodology. Your insight has helped us better articulate the careful reasoning behind our experimental design. We specifically chose the multiple-choice format for PAPI as it aligns with established best practices in personality assessment research. **This standardized approach has been extensively validated in psychological studies and enables reliable, large-scale personality measurement while controlling for response variability.** The IPIP-NEO questionnaires we employed are widely recognized in the field for their robust psychometric properties. Most importantly, this structured format enabled us to **collect and validate an large scale of personality data (>300K samples)**, which would have been extremely challenging with open-ended formats.
>
> Nevertheless, we fully appreciate your concern about response richness and naturalness. This is precisely why we conducted comprehensive **evaluations of open-ended generation** capabilities, thoroughly documented in Section 5.3 and Appendix E: "Open-ended Generation Performance". **Our results demonstrate that models aligned using PAS can successfully generalize from multiple-choice training to produce natural, contextually appropriate open-ended responses.** When evaluated by both GPT-4 and human judges, PAS-aligned models consistently outperformed baselines in generating personality-consistent free-form text (winning rates 41%-45% against various baselines). The human evaluation results are particularly encouraging, showing that the personality traits learned through structured data successfully transfer to natural language generation. We have modified Section 5.3 to highlight this important aspect of our experimental validation.

---

> ### Author Response · Authors · 2024-11-20
> **Response 5**
>
> > The paper also compares PAS to prompting and RL baselines but does not include a comparison to fine-tuning the entire language model.
>
>
>
> Thank you for this valuable suggestion about baseline comparisons. We appreciate the opportunity to clarify our comprehensive evaluation strategy.**We have actually conducted extensive comparisons with full model fine-tuning and other parameter-efficient methods in Appendix E.4. The results are particularly illuminating - as shown in Table 5, PAS demonstrates superior performance across all personality dimensions compared to full fine-tuning and other methods.** Specifically, PAS achieves a composite score of 4.41, outperforming full fine-tuning (4.89), LoRA (4.94), Q-LoRA (5.04), and prompt-tuning (5.15). The performance advantages are consistent across individual traits, with PAS achieving the best scores in Agreeableness (0.94), Conscientiousness (0.91), Extraversion (0.86), and Openness (0.72).
>
>
>
> **Table 5: Performance comparison of different parameter-efficient tuning methods for Llama-3-Instruct 8B**
>
> | Method           | Agreeableness ↓ | Conscientiousness ↓ | Extraversion ↓ | Neuroticism ↓ | Openness ↓ | Composite Score |
> | ---------------- | --------------- | ------------------- | -------------- | ------------- | ---------- | --------------- |
> | Full Fine-tuning | 1.21            | 0.99                | 1.03           | 0.88          | 0.78       | 4.89            |
> | LoRA             | 1.16            | 1.05                | 0.97           | 0.93          | 0.83       | 4.94            |
> | Q-LoRA           | 1.08            | 1.12                | 1.09           | 0.85          | 0.90       | 5.04            |
> | Prompt-tuning    | 1.25            | 1.07                | 1.01           | 0.96          | 0.86       | 5.15            |
> | **PAS (Ours)**   | **0.94**        | **0.91**            | **0.86**       | **0.98**      | **0.72**   | **4.41**        |
>
> **Table 6: Open-ended generation performance comparison for Llama-3-Instruct 8B**
>
> | Method           | PAS Wins | Ties | PAS Loses |
> | ---------------- | -------- | ---- | --------- |
> | Full Fine-tuning | **41%**  | 30%  | 29%       |
> | LoRA             | **43%**  | 33%  | 24%       |
> | Q-LoRA           | **42%**  | 35%  | 23%       |
> | Prompt-tuning    | **45%**  | 31%  | 24%       |
>
>
>
> Even more encouraging are the results from our open-ended generation evaluation (Table 6). **When compared head-to-head in generating natural language responses, PAS consistently demonstrates superior performance against all parameter modification approaches. The win rates are particularly noteworthy: 41% against full fine-tuning, 43% against LoRA, 42% against Q-LoRA, and 45% against prompt-tuning, with relatively low loss rates (23-29%).** These results suggest that PAS's activation-based approach not only matches but exceeds the performance of traditional parameter modification methods, while offering significant advantages in terms of computational efficiency and deployment flexibility. Your comments have helped us better emphasize these important comparative analyses. We have revised Appendix E.4.1.
>
> > Can the authors provide more details on the human evaluation process, such as annotator screening, training, and inter-annotator agreement metrics? This would help validate the human evaluation results.
>
>
>
> Thank you for this important question. We have revised Appendix C.5 to provide comprehensive details about our evaluation process, which was designed to ensure reliability and reproducibility.
>
> Our human evaluation involved three qualified evaluators, all graduate or doctoral students with backgrounds in machine learning, computer science, and cognitive psychology. To establish a robust foundation for evaluation, **each evaluator first completed the IPIP-NEO-120 questionnaire, providing their own personality baseline.** Before beginning the human evaluation, evaluators underwent a standardization process where they reviewed 30 example cases with GPT-4o ratings, ensuring consistent understanding of the evaluation criteria and scoring standards.
>
> The evaluation process itself was systematically structured and controlled. We developed a Python-based annotation tool that presented anonymized samples in randomized order to prevent ordering bias (Figure 14 in the Appendix).  All responses were automatically saved in JSON format for subsequent analysis. **The Inter-Annotator Agreement (IAA) score is 0.82 for human evaluations and 0.85 for human self-evaluators, which validates the reliability and stability of our assessment methodology.** This systematic approach helped ensure consistency and minimize potential biases in the evaluation process. The complete details of our human evaluation protocol, including evaluator selection criteria, training process, and evaluation tools, are documented in Appendix C.5: "Human Evaluation Experiment Details."

---

> ### Author Response · Authors · 2024-11-20
> **Response 6**
>
> > How do you expect the methods to perform on multilingual models and non-English datasets?
>
> The PAPI dataset's inclusion of participants from multiple countries provides a strong foundation for evaluating and validating cross-cultural personality alignment. **PAS operates on internal model activations rather than language-specific features.** This architecture-level intervention approach means PAS can theoretically be applied to any transformer-based language model, regardless of the languages it supports. **Our empirical results support this theoretical advantage - as shown in Table 3 and detailed in Appendix E.5**, PAS demonstrates strong performance across diverse linguistic and cultural groups. For instance, we observe consistently strong alignment scores in non-English speaking countries: China (Openness=1.02, Extraversion=1.06), France (Extraversion=1.17, Openness=1.15), and Germany (Conscientiousness=1.30, Agreeableness=1.38). The robust performance across these linguistically diverse populations suggests that PAS effectively captures personality traits independent of language-specific characteristics.
>
> > The discussion on negative societal impacts of AI hyper-personalization (e.g. filter bubbles, opinion polarization) and is light.
>
> Thank you for raising this critical concern about AI personalization's societal impacts. We have significantly expanded our discussion of ethical implications in a **new "Ethics Statement" section, which thoroughly examines potential risks like psychological filter bubbles and echo chambers.** We particularly focus on how personality-aligned AI systems might inadvertently reinforce existing behavioral patterns, especially for users scoring high in Neuroticism or Dark Triad traits. **The section also addresses privacy concerns and proposes concrete mitigation strategies**, including: (1) a dynamic alignment boundary system that monitors and adjusts alignment intensity to prevent extreme behavioral reinforcement, (2) an adaptive content diversity mechanism that strategically introduces alternative viewpoints while maintaining personality alignment, and (3) robust privacy protection frameworks for securing personality data. We appreciate your comment highlighting these important considerations, as it has helped us develop a more comprehensive framework for responsible deployment of personality-aligned AI systems.
>
> > Finally, exploring beyond multiple choice for collecting preference data, such as open-ended text can be interesting and more useful.
>
> **While open-ended preference data is indeed valuable, we've been particularly mindful of ethical considerations and privacy concerns in collecting large-scale personal narratives. As an exciting alternative approach, we envision leveraging our existing PAPI dataset to generate naturalistic personality descriptions.** For example:
>
> ```markdown
> Original IPIP Responses:
> - "Trust others (Very Accurate)"
> - "Jump into things without thinking (Moderately Inaccurate)"
> - "Dislike yourself (Neither Accurate Nor Inaccurate)"
>
> Potential transformed description:
> "This individual shows a strong tendency to trust and believe in others' good intentions. They typically approach decisions with careful consideration rather than impulsive action. When it comes to self-perception, they maintain a balanced view, neither particularly critical nor overly confident in their self-assessment."
> ```
>
> **This transformation approach could potentially enrich our personality alignment framework while respecting privacy boundaries and ethical constraints inherent in collecting direct personal narratives.** We are excited to explore this direction in future work, as it offers a promising path to combine the reliability of structured assessments with the richness of natural language descriptions.
>
> ---
>
> We hope our comprehensive responses and substantial revisions have adequately addressed all your concerns. Your expert guidance has been invaluable in strengthening this work, and **we would be deeply appreciative if you would kindly reconsider your assessment of our paper in light of these improvements. Thank you again for your time and detailed feedback.**

---

> ### Author Response · Authors · 2024-11-29
>
> Dear Reviewer 4LUh,
>
> As the discussion period is coming to an end soon, we wanted to check if you have had a chance to review our responses. Please let us know if your questions have been adequately addressed - we are happy to provide any additional clarification needed. Thank you for your time!
>
> Best Regards,
>
> Authors of "Personality Alignment of Large Language Models"

---

> ### Author Response · Authors · 2024-12-02
>
> Dear Reviewer 4LUh:
>
> We would like to express our sincere gratitude for your time and effort in reviewing our manuscript and providing valuable feedback. As the deadline for author-reviewer discussions has been extended, we apologize for contacting you again to ensure our responses adequately address your concerns.
>
> A few days ago, we submitted detailed responses to your previous comments, and we sincerely hope these responses effectively address the issues you raised. For example, regarding diversity and data source concerns, we expanded our analysis scope during the discussion phase (including analyses across different ages, genders, and countries, comparing English-speaking vs non-English-speaking countries) and supplemented the Dark Triad inventory analysis and discussion beyond PAPI. Additionally, we revised the fine-tuning baseline comparisons in the appendix and expanded the details of manual evaluation.
>
> We believe these revisions have further strengthened the paper! Please feel free to contact us if you need any clarification or have other questions. We are happy to continue the discussion.
>
> Thank you again, and we look forward to your further feedback.
>
> Best Regards,
>
> Authors of "Personality Alignment of Large Language Models"

---

### Author Response · Authors · 2024-11-20
**General Responses and Summary of Revisions**

We sincerely thank all reviewers for their careful and constructive feedback, which has helped us significantly improve our work. We are encouraged by the reviewers' recognition of several key strengths:

- "A key strength of the work is the PAPI dataset, with over 300,000 real-world...The scale is impressive" (*Review 4LUh*)
- "Overall, I like this paper and I think it's a very good attempt in aligning LLMs from a personality perspective." (*Review 3xm2*)
- "The experimental analysis is comprehensive, with particularly detailed descriptions of the experimental setup." (*Review nZZi*)





Based on your constructive comments, we have made substantial improvements to strengthen our paper's contributions and address the  limitations. The major enhancements include:

1. **Extended Dataset Coverage**: We expanded PAPI with 18,192 Dark Triad questionnaire samples, enabling comprehensive assessment of both positive (Big Five) and negative (Machiavellianism, Narcissism, Psychopathy) personality dimensions. This enhancement provides a more complete framework for personality alignment evaluation (Abstract, Section 1: Introduction, Section 3: Dataset Construction).

2. **Demographic Analysis**: We added comprehensive analysis of PAS performance across different age groups, genders, and nationalities, demonstrating robust alignment capabilities across diverse populations. This validates the broad applicability of our approach (Appendix E.5: Diverse Demographic Groups).

3. **Evaluation Protocol Details**: We supplemented our evaluation framework with specific details about the human evaluation interface, annotator qualifications, and inter-annotator agreement metrics (IAA=0.82). These additions enhance the reproducibility and reliability of our results (Appendix C.5).

4. **Technical Discussion**: We provided additional analysis explaining why scaling laws don't automatically improve personality alignment, highlighting how PAS's targeted intervention achieves better results than larger models through precise control of personality-relevant components (Section 5: Experiments).

5. **Ethics Framework**: We supplemented our discussion with specific mitigation strategies for potential risks of personality alignment, including dynamic alignment boundaries and content diversity systems. This provides a practical framework for responsible deployment (Section 7: Ethics Statement).

6. **Method Analysis Detail**: We added a detailed comparison of how PAS differs from traditional methods in modifying model representations, particularly highlighting our two-stage approach versus global parameter updates. This clarifies PAS's unique advantages in precise personality alignment (Appendix E.6).

With the generous guidance from our esteemed reviewers, we have been able to substantially enhance our paper's technical depth, experimental rigor, and practical impact. We are deeply grateful for the opportunity to improve our work and would be honored if the reviewers find these comprehensive enhancements worthy of a more favorable evaluation. We believe these additions have helped realize the full potential of our research contribution to the field of personality alignment in AI systems.

---

### Author Response · Authors · 2024-12-03

Dear Reviewer 4LUh and nZZi:


Given that the author-reviewer discussion period will **end in an hour**, we are eager to ensure that all your suggestions and comments have been thoroughly addressed!

Once again, thank you for your thoughtful review process, and we look forward to your final comments!

Best Regards,

Authors of "Personality Alignment of Large Language Models"

---

### Meta-Review · Area_Chair_wero · 2024-12-16

**Metareview:**

We recommend the paper to be accepted for Poster.

The paper can be of interest to the wide community at ICLR working on LLM and it introduces a relatively novel methodology that seems to be more efficient than baseline methods.

Below a more detailed description of the paper.

The paper introduces the concept of Personality Alignment, aimed at customizing large language models (LLMs) to align with the preferences and behaviors of individual users or groups. To support this approach, the authors developed a large-scale dataset called PAPI, which contains behavioral preference data from over 300,000 real participants across the Big Five personality dimensions. They also propose a novel method called Personality Activation Search (PAS) to efficiently align LLMs with user preferences during inference. PAS identifies key activation vectors corresponding to personality traits and optimally adjusts activations along these dimensions.

The strengths (S#) of the paper are as follows:

- (S1)	The results demonstrate that PAS outperforms baseline methods in capturing individual preferences while maintaining high computational efficiency
- (S2)	The paper provides a new dataset generation pipeline that can be used and enriched for the task taken into account.
- (S3)	The experimental analysis is comprehensive and provides
- (S4)	The paper is well written, and easy to follow also for a non-specialized audience

The key weaknesses (W#) identified and that remains are as follows:

- (W1)	Use of model weights may hinder the applicability of the method to few LLMs
- (W2)	Possible bias in the annotators used (only from CS community)
- (W3)	Significant disparity between LLM-as-a-judge evaluations and human evaluations should be further discussed

Many of the points raised by the reviewers were addressed by the authors.

**Additional Comments On Reviewer Discussion:**

The authors have been proactive in addressing the comments raised by the reviewers
Reviewer 3xm2 was engaged in reading the authors response and increased the score accordingly, while being confident in the decision.
Reviewers 4LUh and nZZi did not follow up on their reviews after extensive responses from the authors.
As per metareview above, we believe that many of the points raised have been addressed, therefore we lean toward acceptance for poster.

No ethics review raised by the reviewers, and we agree with them.

---

### Decision · Program_Chairs · 2025-01-22

Accept (Poster)